The Company of
Biologists

# Routine metabolic rate is not associated with boldness in zebrafish

Aliyah R. Goldson, Jacob Hudock and Justin W. Kenney*

## ABSTRACT

Consistent individual differences in behavior are prevalent throughout the animal kingdom and are thought to be important contributors to evolutionary adaptation. However, the biological basis for individual differences are poorly understood. One explanatory framework that has gained traction is the pace of life syndrome (POLS) hypothesis. The POLS hypothesis proposes that behavioral variation arises from variation in basic physiological functions like metabolism. In particular, the POLS predicts that individuals with higher baseline metabolic demands will be more willing to take risks to attain the requisite resources. To date, support for this hypothesis when applied within species has been mixed, leading to the tentative conclusion that the relationship between metabolism and behavior depends on factors like species, sex, and context. We sought to determine if the POLS hypothesis held in zebrafish, a widely used model organism with well-developed genetic and neurobiological tools that would allow us to better understand how biological and environmental factors influence predictions of the POLS hypothesis. We tested the POLS hypothesis in adult zebrafish by measuring the relationship between routine metabolic rate, as assessed by oxygen consumption, and exploration of a novel tank. We found no clear relationship between boldness-related behaviors in the novel tank and metabolism in female or male zebrafish of the TU or WIK strains. Our findings suggest that a key prediction of the POLS hypothesis does not hold in zebrafish.

KEY WORDS: Zebrafish, Metabolism, Oxygen consumption, Novel tank test, Boldness, Risk taking

## INTRODUCTION

Individual differences in behavior are observed across many species, contributing to evolutionary adaptation (Dall et al., 2012; Réale et al., 2007; Sommer-Trembo et al., 2024). However, the biological basis for behavioral variation is poorly understood. One proposed explanation for the presence of individual differences comes from the pace of life syndrome (POLS) hypothesis. This hypothesis proposes that variation in innate physiological characteristics, like metabolic rates, drive differences in behavior (Dammhahn et al., 2018; Réale et al., 2010). In particular, the POLS hypothesis proposes that individuals with greater metabolic demands are more willing to take risks to obtain the requisite resources. Despite its intuitive appeal, support for the POLS has been mixed (Royauté et al., 2018); the relationship between metabolic function and behavior turns out to be

Department of Biological Sciences, Wayne State University, Detroit, MI 48202, USA.

*Author for correspondence ( jkenney9@wayne.edu)

(iD) J.H., 0009-0000-6369-9152; J.W.K., 0000-0001-8790-5184

influenced by a complex interplay of species, sex, and context (Dammhahn et al., 2018). For example, there is stronger support for the POLS hypothesis in invertebrates versus vertebrates, and females tend to have correlations in the opposite direction than that predicted by the POLS (Royauté et al., 2018). However, if the relationship between metabolic rate and risk-taking behavior could be established in a widely used model organism, like zebrafish (*Danio rerio*), it would provide a strong platform for developing a better understanding of how physiology influences behavioral variation.

Establishing the POLS hypothesis in zebrafish would enable a better understanding of its mechanistic basis due to the wide variety of genetic and neuroanatomical tools available for this species. Originally developed in the 1980s for developmental biology due to its early life transparency and high fecundity, the past 25 years have seen zebrafish increasingly used to study behavior and its neural basis (Burgess and Burton, 2023; Kenney, 2020). While most of this work has focused on larval animals, over the past decade tools have been developed to enable greater insight from adult animals. Adults have the benefit of more differentiated neuroanatomy and a larger behavioral repertoire than larvae while still being able to make use of the wide array of genetic tools (Hwang et al., 2013; Kawakami, 2004; Marquart et al., 2015). Tools to study behavior and neurobiology in adult zebrafish have proliferated in recent years, including machine learning approaches to automate behavior identification (Fontana et al., 2025; Goodwin et al., 2024; Mathis et al., 2018), efficient capture swimming in three dimensions (Audira et al., 2018; Kuroda, 2018; Maaswinkel et al., 2013; Rajput et al., 2022), refinement of drug delivery (Ochocki and Kenney, 2023), and a digital brain atlas for whole-brain mapping (Kenney et al., 2021; Rajput et al., 2025). Finally, with respect to measuring metabolic function, this is straightforward in adult zebrafish since they consume readily measurable amounts of oxygen while swimming (Cleal et al., 2021).

In the present study, we tested a key prediction from the POLS hypothesis in adult zebrafish: higher metabolic rates are associated with greater risk-taking behavior. To capture metabolic functioning, we measured changes in dissolved oxygen after fish freely swam in a tank (Cleal et al., 2021; Hartman, 2000). We used the novel tank test (NTT) to capture exploratory behaviors associated with risk taking (Maximino et al., 2012; Rajput et al., 2022). We found no correlation between routine metabolic rate (Chabot et al., 2016) and exploratory behavior. To test the POLS hypothesis in a different way, we manipulated metabolic functioning by omitting morning feeding. This resulted in a decrease in oxygen consumption but had little effect on exploratory behavior.

## RESULTS

### Measuring routine metabolic rate

We used changes in dissolved oxygen to assess the routine metabolic rate of adult zebrafish. Fish were placed in a sealed chamber for 30 min and the change in mg of $O_2$/l was measured. Because temperature fluctuations and degassing can alter the amount of dissolved $O_2$ over 30 min in the absence of fish, we compared the change in $O_2$ with fish to the change in $O_2$ in a 'blank tank' that was

run in parallel but did not have a fish. To determine if our approach yielded consistent results, we measured mg of $O_2$/l from the same fish on two consecutive days (Fig. 1A). We found a high level of consistency in our readings using both Pearson's correlations and intraclass correlations (ICC) in both females [r=0.89, P=0.001; ICC=0.80, 95% confidence intervals (CI)=(0.28,0.95)] and males [r=0.94, P<0.001; ICC=0.77, 95% CI=(0.32,0.93)].

To further validate our approach to measuring $O_2$, we examined the relationship between weight and $O_2$ consumption. Based on prior work, we hypothesized that larger fish would consume more $O_2$ (Beamish, 1973; Careau et al., 2008; Clarke and Johnston, 1999; Durnin, 1955). We found a positive relationship between weight and $O_2$ consumption that was more prominent in females than males (Fig. 1B). Females of both the Tübingen (TU) (r=0.50, P=0.005) and Wild India Kolkata (WIK) (ρ=0.52, P=0.002) strains had moderate correlations as did the males (TU: ρ=0.41, P =0.02, WIK: r=0.39, P=0.02). The normality assumption was violated in female WIKs and male TUs, and so Spearman's correlations were used.

Next, we asked if $O_2$ consumption varied by sex and strain (Fig. 1C). To account for weight differences, we normalized $O_2$ consumption by mass to yield a normalized metric (mg $O_2$/l/g of fish). This measure captures the routine metabolic rate, reflecting baseline energy expenditure during spontaneous activity (Chabot et al., 2016; Dall, 1986; Huang et al., 2013; Killen et al., 2021; Polverino et al., 2016). A 2×2 (sex×strain) permutation ANOVA found a small effect of sex (P=0.015, $\eta^2$=0.04) where males had higher routine metabolic rates than females. There was no main effect of strain (P=0.65) or an interaction (P=0.49).

### Routine metabolic rates do not correlate with exploratory behavior

To determine if metabolism was related to the exploratory behavior of zebrafish, we examined the relationship between routine metabolic rate and behavior during exploration of a novel tank. To capture boldness, we calculated a 'boldness index' that combined z-scores for percentage explored and bottom distance (Beigloo et al., 2024;

Rajput et al., 2022). We found no significant correlation between boldness and routine metabolic rate (Fig. 2A, Table S1). We also calculated correlations between the routine metabolic rate and individual behavioral parameters: distance from bottom, distance from center, distance traveled, percent of the tank explored, freezing/immobility, and maximum velocity (Fig. 2B, Table S1). We found only one significant positive correlation between center distance and routine metabolic rate in male WIKs (ρ=0.37, P=0.036); all other correlations were not significant.

### Fasting affects metabolism but not boldness

To further probe a potential metabolism-behavior relationship, we manipulated metabolic rates by omitting the fish's morning feed (Fig. 3A). 2×2 (feed state×sex) permutation ANOVAs within each strain found medium-sized reductions in metabolism due to fasting (TU: P=0.007, $\eta^2$=0.082; WIK: P<0.001, $\eta^2$=0.11). There was also a main effect of sex in TU fish (P<0.001, $\eta^2$=0.13), but not WIKs (P=0.99), where male TUs had a greater metabolic rate than females. There were no interactions between feed state and sex in either TUs (P=0.64) or WIKs (P=0.52).

In a separate cohort of fish, we examined whether fasting affected exploratory behaviors (Fig. 3B-E). There was no effect of fasting on the behaviors most closely associated with boldness: bottom distance (Fig. 3B) or percentage of tank explored (Fig. 3E). However, we did find that fasting increased center distance in both female and male TU fish (P=0.023, d=0.83 and P=0.011, d=0.97, respectively; Fig. 3C). We also found that fasting decreased distance travelled in female WIK fish (P=0.019, d=1.08; Fig. 3D).

### DISCUSSION

We found no relationship between routine metabolic rate and boldness-related exploratory behavior in zebrafish. First, we established that our approach to measuring $O_2$ is consistent across days and correlates with the weight of the fish (Fig. 1). Based on the POLS hypothesis, we expected that animals with greater baseline metabolic activity would be bolder during exploration of a novel

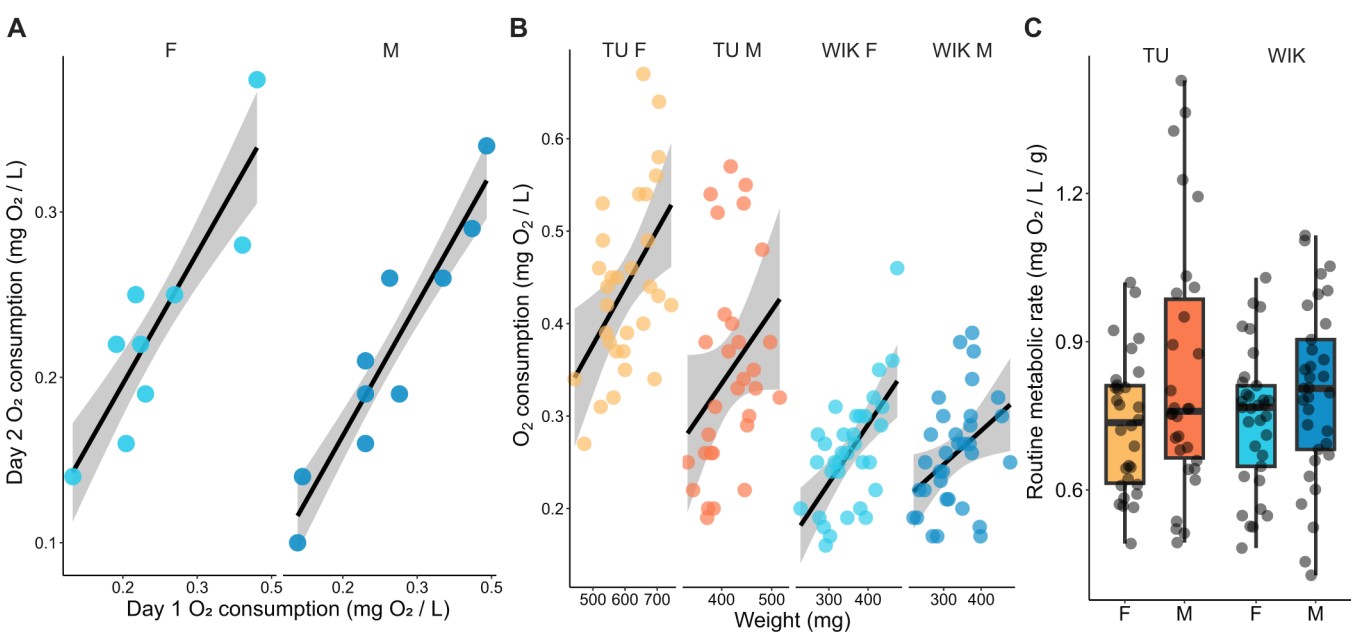

**Fig. 1. Consistency of oxygen measurements and their association with weight, strain, and sex.** (A) Consistency in $O_2$ consumption measured on two consecutive days; female: n=9, male: n=10. (B) Relationship between weight and $O_2$ consumption. (C) Effect of strain and sex on routine metabolic rate. WIK female, n=33; WIK male, n=33; TU female, n=30; WIK male, n=30.

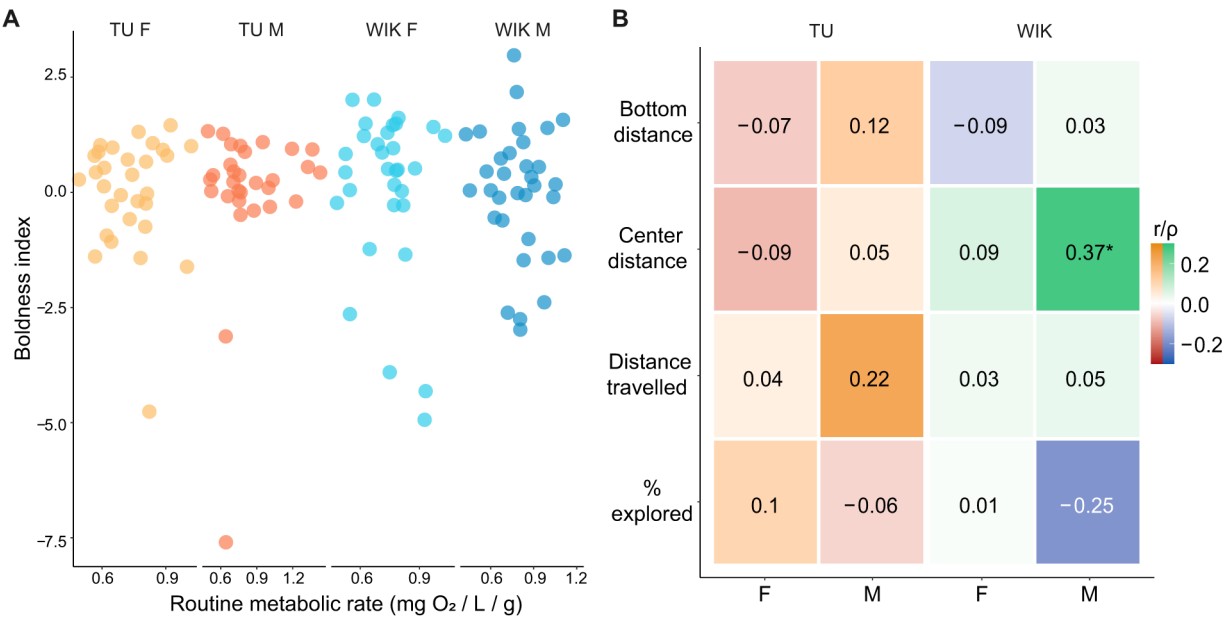

**Fig. 2. Relationship between exploratory behavior and routine metabolic rate.** (A) Relationship between routine metabolic rate and the boldness index. (B) Relationship between routine metabolic rate and individual exploratory behaviors. Correlation type and *P*-values can be found in Table S1. Female TU, *n*=30; male TU, *n*=30; female WIK, *n*=31; male WIK, *n*=33. *\*P*<0.05.

tank. However, we found no such relationship (Fig. 2). We further tested the POLS hypothesis by omitting the morning feed for fish (fasting). While this had the expected effect of reducing the routine metabolic rate, there was minimal impact on exploratory behavior (Fig. 3). Taken together, our study does not provide support for the POLS hypothesis. This is consistent with one prior study in juvenile zebrafish where they found no relationship between routine metabolic rate and behavior in an open field (Polverino et al., 2016).

Prior work testing the predictions of the POLS hypothesis within species have been mixed. Support for and against POLS comes from studies done both in the lab and wild across a wide range of species (reviewed in Royauté et al., 2018). Based on this work, it has been suggested that a variety of factors, like environmental context, genetics, and sex, may affect the potential relationship between metabolism and exploratory behavior (Dammhahn et al., 2018). In the present study, we use inbred zebrafish strains that have been raised in labs for decades. This domestication has resulted in genetic and behavioral differences compared to wild zebrafish (Guryev et al., 2006; Suurväli et al., 2020; Wright et al., 2006). Given the lack of exposure to predators and other risks that would be experienced in the wild, it may be the case that the POLS hypothesis does not hold in such highly domesticated animals. Alternatively, it may be that routine metabolic rate, as measured using O$_2$ consumption, does not capture the most relevant aspect of an individual's metabolic function. For example, Binder and colleagues (2016) found that, in bluegill sunfish (*Lepomis macrochirus*), boldness was associated with maximum, but not basal, metabolic rates. In addition, measuring changes in O$_2$ only captures aerobic metabolism; incorporating anaerobic metabolism would give us a fuller picture (Killen et al., 2015).

We are confident that our results are not due to an inability to properly measure metabolic rate as reflected by changes in dissolved O$_2$. This is for following reasons. (1) The method we used is similar to prior work that successfully measured metabolic function in zebrafish and other fish species (Cleal et al., 2021; Hartman, 2000). (2) Our O$_2$ measurements were highly consistent over 2 days (Fig. 1A), suggesting that we are capturing a stable characteristic of

individual animals. (3) We found a positive correlation between fish weight and routine metabolic rate (Fig. 1B). This last result was anticipated because prior work has found that larger individuals tend to have greater overall metabolic activity (Benton and Hutchins, 2024; Careau et al., 2008; Clarke and Johnston, 1999). Thus, we are confident that we were able to accurately measure the routine metabolic rate of zebrafish and that it represents a consistent characteristic associated with each individual fish.

We found that omitting the morning feed from our fish significantly reduced routine metabolic rate but had little effect on exploratory behavior (Fig. 3). Our findings are similar to that of the recent work from Singh et al. (2025) where they found that hunger in zebrafish did not alter exploratory behavior in an open field, although starvation did decrease emergence time from a T-maze. Likewise, in harvestman spiders (*Mischonyx cuspidatus*), starvation also did not affect boldness (Segovia et al., 2019). However, Eurasian tree sparrows (*Passer montanus*) that undergo fasting do increase their exploratory behavior (Lee et al., 2016). This variation in results suggests that the relationship between food deprivation and behavior likely depends on factors like species, environmental context (e.g. wild versus the lab), and the specific measure used for boldness.

The only relationship between behavior and metabolic functioning we found was with center distance [a correlation between center-distance and routine metabolic rate in male WIK fish (Fig. 2B) and a fasting-induced increase in center distance in TU fish (Fig. 3C)]. Increased center distance in zebrafish is typically interpreted as an increase in anxiety-related or shy behavior based on its similarity to thigmotaxis in rodents (Borba et al., 2025; Champagne et al., 2010; Kalueff et al., 2013). Thus, one interpretation of our findings is that the reduction in metabolic activity due to fasting results in a strain specific decrease in thigmotaxis-related boldness, which is consistent with the POLS hypothesis. However, the correlation we observed in male WIK fish is in the opposite direction than what the POLS would predict (i.e. higher routine metabolic rate is associated with more time in the periphery) (Fig. 2B). Complicating the interpretation is that the effects of fasting on behavior may

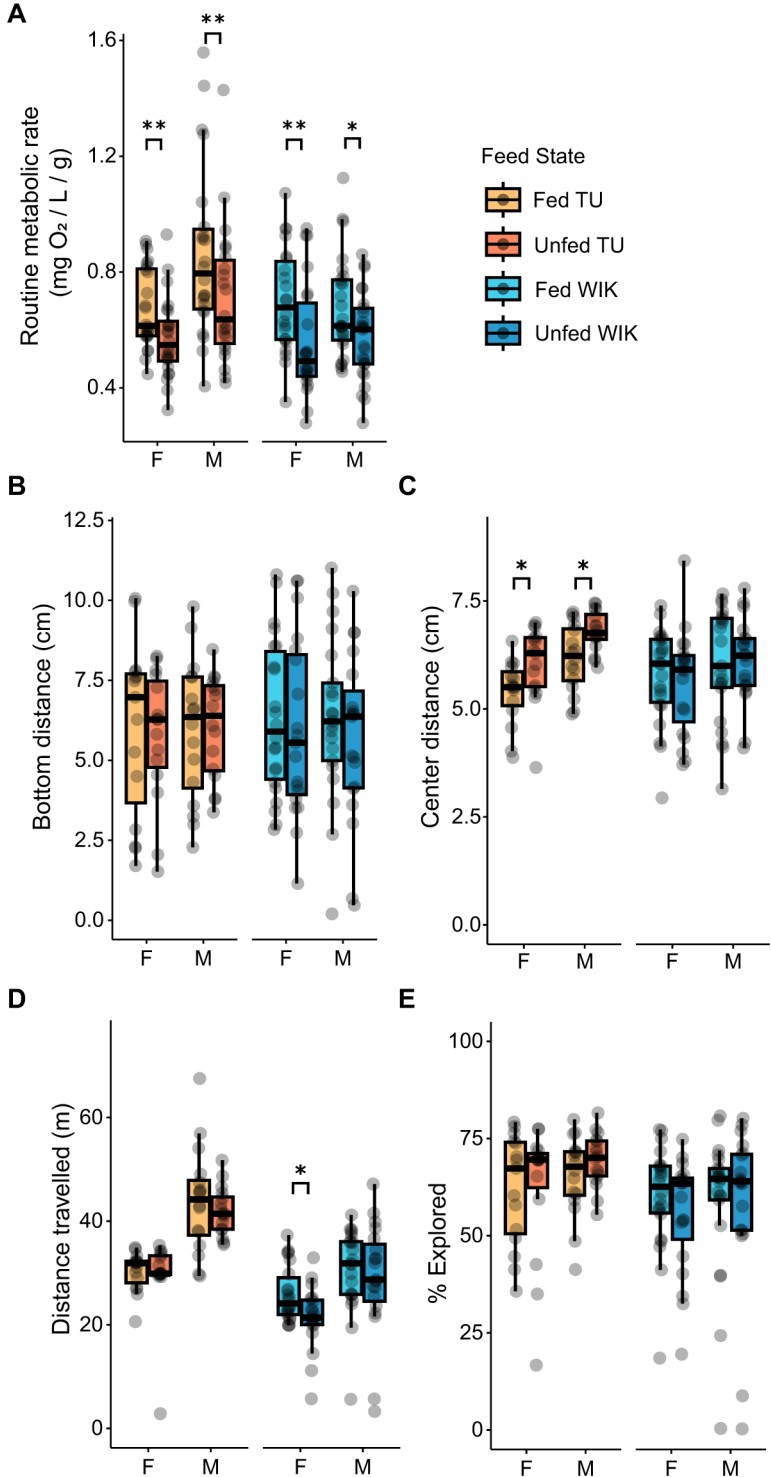

**Fig. 3. The effects of 16-h fasting on routine metabolic rate and behavior.** (A) The effect of fasting on routine metabolic rates. Paired Wilcoxon rank sum tests within sex and strain. Female TU ($P=0.0008$; $n=22$), male TU ($P=0.001$, $n=24$), female WIK ($P=0.002$; $n=23$), male WIK ($P=0.037$; $n=22$). (B) The effects of fasting on bottom distance. Female TU ($P=0.96$), male TU ($P=0.89$), female WIK ($P=0.35$), male WIK ($P=0.27$). (C) The effects of fasting on center distance. Female TU ($P=0.022$§), male TU ($P=0.011$), female WIK ($P=0.97$), male WIK ($P=0.27$). (D) The effect of fasting on distance travelled. Female TU ($P=0.74$§), male TU ($P=0.50$), female WIK ($P=0.019$), male WIK ($P=0.55$). (E) The effect of fasting on percent explored. Female TU ($P=0.66$§), male TU ($P=0.22$), female WIK ($P=0.16$§), male WIK ($P=0.92$§). TU female fed, $n=15$; TU female unfed, $n=15$; TU male fed, $n=16$; TU male unfed, $n=16$; WIK female fed, $n=22$; WIK female unfed, $n=19$; WIK male fed, $n=21$; WIK male unfed, $n=19$. *$P<0.05$, **$P<0.01$. §, Mann–Whitney $U$ test was used.

not be mediated by the alteration in metabolic functioning but instead may be working via other mechanisms like hunger or food seeking (Filosa et al., 2016; Nakajo et al., 2020; Singh et al., 2025). Furthermore, the interpretation of center distance in zebrafish is not entirely clear given that it does not consistently relate to other more commonly used measures of predator avoidance or anxiety-like behavior in zebrafish (Blaser et al., 2010; Borba et al., 2025; Champagne et al., 2010; Rajput et al., 2022; Rosa et al., 2018; Wong et al., 2012). Thus, further work is needed to clearly interpret these results.

Identifying the underpinnings of individual differences in behavior remains an important challenge in neuroscience, precision medicine, and behavioral ecology. Our findings from domesticated zebrafish do not clearly support the POLS hypothesis. Thus, other organisms or contexts (e.g. in the wild or under more explicit predation risk) are likely needed to evaluate potential mechanistic links between metabolic functioning and behavior. Our findings also suggest that it is likely other factors, beyond metabolic functioning, that drive individual variation in adult zebrafish behavior.

## MATERIALS AND METHODS

### Subjects

Subjects were TU and WIK zebrafish aged 4 to 8 months. The TU strain was established from a pet shop in the 1990s and was bred to remove embryonic lethal mutations (Haffter et al., 1996; Mullins et al., 1994) and used to generate the zebrafish genome (Howe et al., 2013). In contrast, the WIK line was more recently established from wild caught fish. They are more genetically polymorphic than the TU line (Rauch et al., 1997), more closely resembling wild animals (Wilson et al., 2014). All fish were bred and raised at Wayne State University and within two generations of animals obtained from the Zebrafish International Resource Center at the University of Oregon. Animals were kept on high density racks (20-40 fish per 4 or 8 l tanks at a density of ∼5 fish/l) under standard conditions (temperature 27.5±0.5°C; water conductivity 500±10 μS, and a pH of 7.5±0.2) with a 14:10 light:dark cycle (lights on at 08:00). Fish were fed twice a day with a dry feed in the morning (Gemma 300, Skretting, Westbrook, ME, USA) and brine shrimp (Artemia salina; Brine Shrimp Direct, Ogden, UT, USA) in the afternoon. Behavioral testing took place between 11:00 and 14:00. The sex of fish was determined using three secondary sexual characteristics: shape (prominent belly for females), color (with males being more red/pink), and presence of pectoral fin tubercles (exclusively found on males; McMillan et al., 2015). We dissected animals after experiments to confirm the presence or absence of eggs. Animals that were assigned the wrong sex were removed from analysis. One fish with negative oxygen consumption was removed from analysis as this was likely due to a data entry error. All procedures were approved by the Wayne State University Institutional Animal Care and Use Committee.

### Novel tank test

The novel tanks consisted of five-sided tanks (15×15×15 cm) made from frosted acrylic (TAP Plastics, San Leandro, CA, USA). Tanks were placed in an enclosure of white corrugated plastic to diffuse light and prevent exposure to external stimuli. D435 Intel RealSense™ cameras (Intel, Santa Clara, CA, USA) were mounted 20 cm above tanks to capture three-dimensional videos (Kuroda, 2018; Rajput et al., 2022). These cameras capture three-dimensional videos using the synchronous capture of two video streams: a color stream (red/green/blue) and a depth stream. The depth stream is generated via stereoscopic imaging using the disparity between two infrared cameras. Firmware on the camera synchronizes the capture of the two streams. Cameras were connected to a Linux workstation via high-speed USB cables (NTC Distributing, Santa Clara, CA, USA), and video capture was controlled via custom written Python scripts. Animals with videos that were not fully recorded due to malfunction were not analyzed. Experimental tanks were filled with 2.5 l of fish facility water. Individual fish were placed in the tanks for 6 min while video was recorded for offline analysis. Tanks were rinsed between animals and water was replaced.

One week prior to testing, fish were placed as male/female pairs into 2-l tanks to enable non-invasive identification across days. The tanks were divided in half using a transparent divider with two fish in each section and four fish in each tank. On days when behavior was assessed, fish were taken off housing racks and moved to the procedural space at least 1 h prior to testing. Following testing, fish waited at least 30 min before being returned to the housing racks.

### Animal tracking and behavioral analysis

Fish were tracked along five points on the body using DeepLabCut (Mathis et al., 2018). The model was trained as previously described (Rajput et al., 2022). Animals unable to be accurately tracked were removed from analysis (one fish). After tracking, four parameters were extracted: distance from tank center, distance travelled, distance from bottom, and percentage explored, wherein the tank was divided into 1000 voxels, and the percentage of explored voxels was calculated.

We generated a boldness index that combined the z-scores for the percentage of the tank explored and the bottom distance. These were calculated within each sex and strain. The boldness index is based on prior work that found that these two measures best distinguished bold and shy fish (Beigloo et al., 2024; Rajput et al., 2022).

We also calculated the percentage of time fish spent immobile and their maximum velocity. For immobility, we first plotted a distribution of velocities and found a break in the data at ∼1.5 mm/s (Fig. S1), which we used as a cut-off for determining immobility. For maximum velocity, we used the median of the top 5% of velocity measurements for each fish.

### Dissolved O$_2$ measurements

Dissolved O$_2$ measurements were taken while fish swam in custom built tanks [15.2 cm (L)×8.9 cm (W)×6.0 cm (H)] made from P95 clear and frosted acrylic (4.8 mm thick; TAP Plastics). Each tank was filled with 500 ml of facility water and placed in the same enclosure used for novel tank tests. The tank lid consisted of two layers: (1) a top layer of P99 non-glare acrylic (3.2 mm thick) and (2) a bottom layer of clear cast acrylic (4.8 mm), sealed with a bead of aquarium silicone (Aqueon, Franklin, WI, USA) along the rim to ensure an air-tight seal.

Dissolved O$_2$ was measured using an optical meter (DO850, Apera Instruments, Columbus, OH, USA) as mg of O$_2$/l. We measured the initial and final O$_2$ concentrations before and after fish were placed in the sealed tank for 30 min. To account for changes in dissolved oxygen unrelated to fish metabolism, such as degassing or temperature changes, oxygen levels in a 'blank tank' containing the same volume of water, but no fish, were simultaneously recorded. This change in dissolved oxygen of the 'blank tank' was subtracted from that of the tank containing the fish to obtain each fish's overall oxygen consumption:

$$[O_2]_{consumed} = ([O_2]_{f,t=1} - [O_2]_{f,t=2}) - ([O_2]_{b,t=1} - [O_2]_{b,t=2}),$$

where $[O_2]_{consumed}$ is the amount of oxygen consumed by the fish, $[O_2]_f$ is oxygen measured in the chamber containing the fish, $[O_2]_b$ is the oxygen measured in the 'blank tank', and $t=1,2$ corresponds to the beginning and end of the trial, respectively.

### 16-h fasting

Male and female fish were randomly assigned to either the fasted ('unfed') or control ('fed') groups prior to the experiment. Fish were fasted for approximately 16 h by withholding their morning feed. Control groups were fed their morning feed approximately 1 h prior to the experiment. 15 min after feed administration, fish were transported to the experimental room and left to habituate for 1 h. Individual fish of each group were then placed into the dissolved oxygen chamber for 30 min to capture their metabolic activity. 2 days later, individuals were switched between groups and oxygen consumption was measured. A separate cohort of fish were divided into fed and unfed groups and tested in the novel tank as described above.

### Coding and statistical analysis

Data analysis was performed using R version 4.5.0 (https://www.r-project.org/; https://cir.nii.ac.jp/crid/1574231874043578752). All graphs were generated using ggplot2 (Wickham, 2016). Normality was assessed using the Shapiro-Wilks test. For correlations, we used Pearson's correlations if the normality assumption was not violated otherwise, we used Spearman's ρ. To determine repeatability, we used ICC and calculated 95% CI by bootstrapping 1000 times with replacement using the rptr package (version 0.9.23). For comparing four groups we used permutation ANOVAs (with 10,000 resamples). For comparing two groups, we used Welsch's independent sample $t$-tests, unless the normality assumption was violated, then we used the Mann–Whitney $U$ test for independent samples or Wilcoxon rank-sum tests for paired samples. Effect sizes were calculated using Cohen's d for two groups or $\eta^2$ for ANOVAs. Interpretation of effect sizes were small ($0.01 < \eta^2 < 0.06$; $0.2 < d < 0.5$), medium ($0.06 \leq \eta^2 < 0.14$; $0.5 \leq d < 0.8$) or large ($\eta^2 \geq 0.14$; $d \geq 0.8$) based on Cohen (1988). Sample sizes were chosen to be able to detect medium sized effects based on estimations of our prior work with behavior in adult zebrafish. All experiments were performed once in the lab.

### Acknowledgements

We thank Dinh Luong for excellent zebrafish care.

## Competing interests
The authors declare no competing or financial interests.

## Author contributions
Conceptualization: A.R.G., J.H., J.W.K.; Data curation: J.W.K.; Formal analysis: A.R.G., J.W.K.; Funding acquisition: J.W.K.; Investigation: A.R.G.; Methodology: A.R.G., J.H., J.W.K.; Project administration: J.W.K.; Resources: J.W.K.; Supervision: J.W.K.; Visualization: A.R.G., J.W.K.; Writing – original draft: A.R.G., J.W.K.; Writing – review & editing: A.R.G., J.W.K.

## Funding
This work was funded by the National Institute of General Medical Science (R35GM142566 to J.W.K.). Open Access funding provided by Wayne State University. Deposited in PMC for immediate release.

## Data and resource availability
All relevant data and details of resources can be found within the article and its supplementary information. Data are also available from the Dryad repository (https://doi.org/10.5061/dryad.83bk3jb6b).

## Peer review history
The peer review history is available online at https://journals.biologists.com/bio/lookup/doi/10.1242/bio.062329.reviewer-comments.pdf

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
