## [Peer Review File · Biology Open]

Routine metabolic rate is not associated with boldness in zebrafish

Aliyah R. Goldson, Jacob Hudock and Justin W. Kenney

DOI: 10.1242/bio.062329

Editor: Lewis Halsey

Review timeline

Original submission: 19 October 2025

Editorial decision: 26 October 2025

First revision received: 24 February 2026

Accepted: 26 February 2026

Original submission

First decision letter

MS ID#: bio.062329

MS Title: Routine metabolic rate is not associated with boldness in zebrafish

Authors: Aliyah R Goldson; Jacob Hudock; Justin W Kenney

I have now reached a decision on the above manuscript.

The reviewer reports are shown at the bottom of this email or can be accessed, together with a copy of this decision letter, by going to:

As you will see, the reviewers raised a number of substantial criticisms that prevent me from accepting the paper at this stage.

They suggest, however, that a revised version might prove acceptable, if you can address their concerns. If you think that you can deal satisfactorily with the criticisms on revision, I would be pleased to see a revised manuscript. We would then return it to the reviewers.

At this stage, we also ask you to ensure your manuscript complies with our formatting guidelines. Provided you are able to fully address the referees' comments, we are positive about publication of your paper (we accept over 95% of revision submissions) and therefore hope you won't mind any extra work involved in reformatting your manuscript at this point.

Please upload both a 'clean' version of your Word file, along with a highlighted version clearly showing where you have made changes in the revised manuscript. Please avoid using 'Track changes' in Word files as these are lost in PDF conversion.

I should be grateful if you would also provide a point-by-point response detailing how you have dealt with the points raised by the reviewers in the 'Response to Reviewers' box. Please attend to all of the reviewers' comments. If you do not agree with any of their criticisms or suggestions please explain clearly why this is so.

Reviewer 1

Comments for the author

Overall comments: In the attached manuscript, Goldson et al test the hypothesis that metabolism correlates with boldness behaviour in zebrafish through measurements of both metabolic rate (using aquatic respirometry) and behaviour (video tracking). This test of the POLS hypothesis is creative and useful. This is one of those studies for which even negative results are useful to the field, particularly as zebrafish are so widely used...developing a behavioural model of the POLS hypothesis in this organism would have been a powerful development but demonstrating that this relationship is not present is also valuable. Overall, the manuscript is very well-written and reasoned. The justification for the experiments is clearly laid out in the introduction, and the hypothesis is clear. The methods and statistical analysis are rigorous and include consideration of sex and strain. Results and statistical evaluations are clearly presented. The figures are appropriate for the data presented and the figure legends are clear and make it easy to understand the results. The discussion of the results is reasonable and measured. Overall, I have only a few relatively minor comments for the authors to consider in revising their submission.

Minor Comments:

1. The authors might consider analyzing time spent active and velocity as well as their present measures of boldness and total distance moved. These measures would give them an indication of whether there are differences in burst or more intense activity in some fish vs. others.
2. The interpretation of the respirometry data is reasonable but of course measuring dissolved O₂ consumption only measures aerobic metabolism. It might be worth noting in the discussion that changes in anaerobic metabolism may occur that are not measured and thus there may still be a masked relationship between metabolic rate and boldness if total metabolic turnover is considered.

Reviewer 2

Comments for the author

Goldson and colleagues conducted a study to test the pace of life syndrome hypothesis using inbred zebrafish strains, specifically looking at whether there was a relationship between resting metabolic rate and boldness. They found that while there was strong correlation between metabolic rate over time, resting metabolic rate was not related to boldness in both strains of zebrafish they tested. The study design was generally appropriate for investigation of the research question, and the study setup was well-described in the manuscript. I do however have some general questions and suggestions, especially regarding the novelty of the study and the choice of using two strains of zebrafish. The authors mentioned in the introduction that the POLS hypothesis has been tested in many organisms and that "the relationship between metabolic function and behavior turns out to be influenced by a complex interplay of species, sex, and context". While I am not dismissing the value of this study, I wonder if the authors could specify what the main novelty is and how the study could address the gap of knowledge and truly contribute to our understanding of POLS. As a non-zebrafish person, I also wonder about the choice of using the two strains of zebrafish for this study. What is the rationale for using two different strains and why use specifically the TU and WIK strains? Specific comments below.

Specific comments:

Line 43-46: This statement could be developed further. I would appreciate more elaboration and examples of evidence for and against POLS and behaviour.

Line 46-49: How is a widely-used model system superior to non-model organisms for studying this question? One could also argue that most lab-bred model systems have limited genetic diversity and thus offer little physiological and behavioural variations for investigation of this question.

Line 66-68: I struggle to understand the true novelty of this study if it only adds another species to the long list of species that other people have studied in relation to POLS. The authors already noted earlier that "the relationship between metabolic function and behavior turns out to be

influenced by a complex interplay of species, sex, and context". What exactly is unique about the current study and model system that allows you to test the POLS hypothesis?

Line 71-72: What is the purpose of metabolic rate manipulation? Why is it relevant for this study? It is only implied, but I suggest stating it explicitly.

Line 75-76: I suggest removing this sentence from the introduction section as it should belong to the result and/or conclusion sections.

Line 80: Please describe the two strains. What is the rationale for using two different strains and are there any differences between them?

Line 82-85: How many fish are in each tank?

Line 135: These chambers are quite big for a zebrafish, so presumably animals are able to move around during measurement. Is that the case? And if so, perhaps resting metabolic rate is not the appropriate term to use for this study.

Line 164-170: This section needs to be developed further. I suggest that the authors describe the statistics used for each of the parameters analysed, rather than just listing all the statistical tests. Did you check for normality and homogeneity of variance in your data?

Line 194-196: Correlation is not the best test for repeatability in my opinion. I suggest calculating intra-class correlation coefficients to check how repeatable the metabolic rates are between two days. What about repeatability of behaviour? Are measures of boldness repeatable over time?

Line 213-222: There is a bit of inconsistency in terms of how the results are presented here. In the other subsections, the statistics are listed but not in this subsection.

Line 256-261: Are the variations observed in metabolic rate and behaviour of inbred zebrafish similar to variations observed in wild zebra fish? Could the absence of correlation between metabolic rate and boldness in inbred zebra fish be attributed to limited individual variations rather than simply a genetic or behaviour difference compared to wild population?

Line 269-271: See previous comment on repeatability testing.

Reviewer's Responses to Questions

Experimental quality

Does each figure have the proper controls?

If 'No', please indicate reasons in Comments for Author box below.

Reviewer #1:

- Yes

Reviewer #2:

- Yes

Were the data analyzed using appropriate statistical tests?

If 'No', please indicate reasons in Comments for Author box below.

Reviewer #1:

- Yes

Reviewer #2:

- No

Reproducibility

Were experiments performed using adequate number of biological replicates?
If 'No', please indicate reasons in Comments for Author box below.

Reviewer #1:

- Yes

Reviewer #2:

- Yes

Does the methods section provide sufficient detail to permit reproducibility?
If 'No', please indicate reasons in Comments for Author box below.

Reviewer #1:

- Yes

Reviewer #2:

- Yes

Completeness

Are the manuscript's conclusions supported by the data?
If 'No', please indicate reasons in Comments for Author box below.

Reviewer #1:

- Yes

Reviewer #2:

- Yes

Scholarship

Do the authors cite and discuss the merits of data that would argue for and against their conclusion?
If 'No', please indicate reasons in Comments for Author box below.

Reviewer #1:

- Yes

Reviewer #2:

- Yes

Does the manuscript title & abstract accurately reflect the contents of the manuscript, without hyperbole?
If 'No', please indicate reasons in Comments for Author box below.

Reviewer #1:

- Yes

Reviewer #2:

- Yes
-

First revision

Author response to reviewers' comments

Thank you for sending our manuscript entitled “Resting metabolic rate is not associated with boldness in zebrafish” out for review. We were pleased to see that the reviewers considered the manuscript ‘well-written and reasoned’ with a discussion that is ‘reasonable and measured’ (reviewer 1). In addition, reviewer 2 remarked that the ‘study design was generally appropriate for the investigation’ and ‘the study setup was well-described’.

We also appreciate the constructive feedback from the reviewers that has improved the manuscript. Below, we outline our response (in italics) to each of the reviewer’s points.

Reviewer #1

“The authors might consider analyzing time spent active and velocity as well as their present measures of boldness and total distance moved. These measures would give them an indication of whether there are differences in burst or more intense activity in some fish vs. others.”

We have now added this additional analysis (*lines 159-163; 269-271; Table S1*). We did not find any relationship between resting metabolic rate and time spent active or the maximum velocity of fish (median of top 5% velocities during swimming).

“The interpretation of the respirometry data is reasonable but of course measuring dissolved O₂ consumption only measures aerobic metabolism. It might be worth noting in the discussion that changes in anaerobic metabolism may occur that are not measured and thus there may still be a masked relationship between metabolic rate and boldness if total metabolic turnover is considered.”

Thank you for this suggestion, it is an excellent point. We have now added this additional information to the discussion (*lines 316-318*).

Reviewer #2

Line 43-46: This statement could be developed further. I would appreciate more elaboration and examples of evidence for and against POLS and behaviour.

We have now added more information to this section, noting that the POLS hypothesis has more support in invertebrates than vertebrates, and that females tend to have a trend in the opposite direction (*lines 46-48*).

Line 46-49: How is a widely-used model system superior to non-model organisms for studying this question? One could also argue that most lab-bred model systems have limited genetic diversity and thus offer little physiological and behavioural variations for investigation of this question.

Thank you for bringing this point up; we have now re-worded the introduction to make it clearer as to why we believe it is worthwhile to explore the POLS in a widely-used model organism like zebrafish. In short, the strength of working with a lab-bred model organism is that we have greater availability of tools that can give us deeper mechanistic insight. For example, we have a digital brain atlas for adult zebrafish, a tool that is lacking in the vast majority of organisms. Such a tool is key for generating unbiased insight into brain function.

Our improved discussion of this point is on *lines 53-55, lines 61-63* and has been added to the abstract (*lines 23-26*) so this idea is clearer to readers.

Line 66-68: I struggle to understand the true novelty of this study if it only adds another species to the long list of species that other people have studied in relation to POLS. The authors already

noted earlier that "the relationship between metabolic function and behavior turns out to be influenced by a complex interplay of species, sex, and context". What exactly is unique about the current study and model system that allows you to test the POLS hypothesis?

We agree with the reviewer that this study is not particularly novel beyond testing the POLS in another species, zebrafish. However, the negative finding here is important because it suggests that this widely used model organism is not useful for understanding the POLS. This negative data will help inform other researchers if it is worth their time and resources to explore the POLS in zebrafish. Alternatively, if we had found support for the POLS in zebrafish, this would have opened the door to better understanding the mechanistic basis for the POLS because of the wide variety of genetic and neuroanatomical tools available for this species (as noted in our response to the previous question).

Line 71-72: What is the purpose of metabolic rate manipulation? Why is it relevant for this study? It is only implied, but I suggest stating it explicitly.

The purpose of this experiment is to test the POLS in a different way, potentially revealing conditions under which the POLS might hold despite the lack of correlation between metabolic functioning and behavior. We believe testing different predictions of the POLS makes our work stronger and highlights the strength of using a model system (i.e., we're able to manipulate things like feeding easily). We have now added lines to the manuscript making this point (**lines 77-78**)

Line 75-76: I suggest removing this sentence from the introduction section as it should belong to the result and/or conclusion sections.

Thank you for this suggestion. We agree and have now removed this line.

Line 80: Please describe the two strains. What is the rationale for using two different strains and are there any differences between them?

Thank you for this suggestion; the addition of this information strengthens the manuscript, especially for readers not as familiar with zebrafish as a model organism. In short, the TU line is more inbred, but have been widely used since they were the strain used for generating the zebrafish genome. The WIK line is more genetically polymorphic and more closely related to wild fish (**lines 102-108**).

Line 82-85: How many fish are in each tank?

We raise our fish at ~5 fish per liter in 4 or 8 liter tanks (20-40 fish per tank). We have now added this information to the methods (**lines 110-111**).

Line 135: These chambers are quite big for a zebrafish, so presumably animals are able to move around during measurement. Is that the case? And if so, perhaps resting metabolic rate is not the appropriate term to use for this study.

Thank you for prompting us to think more about the terminology here. We've decided to change the terminology we use to 'routine metabolic rate' based on our reading of Chabot et al (2016). This captures the idea of spontaneous movements without assuming that fish are static in the tanks. We have now updated this throughout the manuscript.

Reference

Chabot, D., Steffensen, J. F., & Farrell, A. P. (2016). The determination of standard metabolic rate in fishes. *Journal of Fish Biology*, 88(1), 81-121.

Line 164-170: This section needs to be developed further. I suggest that the authors describe the statistics used for each of the parameters analysed, rather than just listing all the statistical tests. Did you check for normality and homogeneity of variance in your data?

We thank the reviewer for pushing us to clarify our statistics a bit more. For ANOVAs, we now use permutation ANOVAs that do not require the normality assumption to be met (the results were

nearly identical).

All t-tests were Welsch's t-tests that do not assume homogeneity of variance. We now added this point to the methods section.

We have also added the use of the Shapiro-Wilks test to test normality and if we found the assumption violated, we used Spearman's correlations or the Mann-Whitney U test as appropriate. Our updated methods now reflect these changes (*lines 197-207*). To help the reader decipher the correlation in figure 2B, we have added a supplemental table (Table S1) that indicates if Pearson's or Spearman's correlations were used.

Finally, instead of listing which test was used for each experiment here, we make sure to indicate the statistical test in either the text of the results, the supplemental table (Table S1) or the figure captions.

Line 194-196: Correlation is not the best test for repeatability in my opinion. I suggest calculating intra- class correlation coefficients to check how repeatable the metabolic rates are between two days. What about repeatability of behaviour? Are measures of boldness repeatable over time?

We have now included intra-class correlations in addition to the Pearson's correlations we previously reported (*lines 238-241*). In short, the intra-class correlations also suggest our measurements of DO show good reliability (0.77-0.80).

As for the repeatability for measures of boldness, we did not measure that here, however, we have in prior work finding it to be repeatable over time (Rajput et al, 2022). However, we would like to note, it is challenging to measure the consistency of boldness over time as habituation inevitably occurs as fish get exposed to the novel environment. Thus, in general, we consider the behavior on the first day of exposure to the environment to be the best indicator of boldness.

Line 213-222: There is a bit of inconsistency in terms of how the results are presented here. In the other subsections, the statistics are listed but not in this subsection.

We only include significant effects in the written results to increase readability. However, to help the reader better interpret this section we have now included a supplemental table (Table S1) that includes all the correlation and p-values for this section.

Line 256-261: Are the variations observed in metabolic rate and behaviour of inbred zebrafish similar to variations observed in wild zebra fish? Could the absence of correlation between metabolic rate and boldness in inbred zebra fish be attributed to limited individual variations rather than simply a genetic or behaviour difference compared to wild population?

This is a great question for which we do not have an answer. It could very well be that we do not see a relationship between metabolism and behavior because we are working with highly domesticated lines and not wild zebrafish. However, we are not in a position to be able to answer this question. Nonetheless, we do discuss this caveat in the discussion (*lines 280-287*)

Line 269-271: See previous comment on repeatability testing.

We have addressed this issue above by using the intra-class correlation coefficient that suggests our readings are reliable across days (*lines 213-215*).

Second decision letter

MS ID#: bio.062329R1

MS Title: Routine metabolic rate is not associated with boldness in zebrafish

Authors: Aliyah R Goldson; Jacob Hudock; Justin W Kenney

I have had time today to read through your responses to the two referees along with the associated edits to your submission, and I am happy to tell you that your manuscript has been accepted for publication in Biology Open, pending our standard publication integrity checks. It was accepted on 26th February 2026.